

# The current sheet flapping motions induced by non-adiabatic ions: case study

Xinhua Wei[1], Chunlin Cai[1], Henri Rème[2], Iannis Dandouras[2], George Parks[3]

1 State Key Laboratory of Space Weather, National Space Science Center, CAS, Beijing, China

2 IRAP, UPS-OMP, Université de Toulouse, CNRS, Toulouse, France

3 Space Sciences Laboratory, University of California, Berkeley, California, USA



## Abstract

In this paper, we analyzed the y-component of magnetic field line curvature in the

plasma sheet and found that there are two kinds of shear structures of the flapping

current sheet, i.e. symmetric and antisymmetric. The alternating bending orientations

of guiding field are exactly corresponding to alternating north-south asymmetries of

the bouncing ion population in the sheet center. Those alternating asymmetric plasma

sources consequently induce the current sheet flapping motion as a driver. In addition,

a substantial particle population with dawnward motion was observed in the center of

a bifurcated current sheet. This population is identified as the quasi-adiabatic particles,

and provides a net current opposite to the conventional cross-tail current.



## 1. Introduction


The magnetotail current sheet, which separates the northern lobe of the magnetotail
from the southern lobe, is one of key objects of magnetospheric physics. As early as
in 1967, the magnetotail current sheet was observed to move in the north-south
direction [Speiser and Ness, 1967], which has been referred to as flapping motions
[Lui et al., 1978; Sergeev et al., 1998]. Previous studies based on single-spacecraft
measurements show that the geometry of the flapping current sheet is more complex
than a planar surface, and that the sheet may be wavy [Lui et al.,1978; Nakagawa and
Nishida,1989].
The multi-point Cluster spacecraft have provided the chance to distinguish the spatial
and time resolutions of the current sheet flapping. In the past two decades, intense
investigations using Cluster data were performed to analyze the magnetic and current
structures of the flapping current sheet. The flapping motion of the current sheet is
interpreted as large amplitude waves with periods from about 30s to several minutes
and amplitudes from several nT to dozens of nT [Zhang et al., 2005, 2006; Runov et
al., 2003, 2005; Sergeev et al., 2003; Cai et al., 2008]. The flapping wave propagates
from the central sector of the magnetotail toward the flanks [Sergeev et al, 2004], and
the propagation velocities of waves are in the range of several tens km/s for the
locally quiet sheets, and up to 200km/s during fast flows. Runov et al. [2005]
performed a statistical analysis of the electric current and magnetic field geometries of
flapping magnetotail current sheets during the period from July to October 2001.
Their results show that $J_z$ is often larger than $J_y$ and the current is almost vertical, and



the flapping current sheet strongly deviates from the nominal plane geometry. The
flapping waves can extend over ~10 $R_e$ in the sun-earth direction by comparing the
data from the Cluster and Double Star missions [Zhang et al, 2005]. In addition, the
joint observations of THEMIS and Cluster show that the flapping amplitude of the
current sheet is from 1 to 3 $R_e$ [Runov et al, 2009], and the flapping waves are steep
tail-aligned structures with a longitudinal scale of >10 $R_e$.
The triggering mechanism of current sheet flapping looks rather mysterious. Early
researchers suggest that flapping motions of the tail plasma sheet can be induced by
the interplanetary magnetic field variations [Toichi Tsutomu and Miyazaki Teruki
1976]. The solar wind dynamic pressure pulse and substorm are also considered to be
the sources of flapping waves [Forsyth et al., 2009]. Some current sheet flapping
events were often observed in association with magnetospheric activities. In 1976,
Toichi and Miyazaki found that current sheet flapping motions occur during the early
phases of substorms. Sergeev et al [1998] showed that current sheet flapping events
are observed around substorm onsets while Sergeev et al. [2006], utilizing the Geotail
data, showed that the majority of fast crossings of the current sheet occurred during
the period of low magnetic activities. Thus, the relationship of the substorm and the
current sheet flapping is still unclear. The flapping wavy motions can also be
generated by the Kelvin-Helmholtz instability in the presence of dawn-dusk flows
with a speed of several tens of km/s are present [Nakagawa and Nishida, 1989].
Furthermore, when hot magnetospheric plasma is confined in a curvilinear magnetic
field, the ballooning mode instability is also able to excite the flapping motion

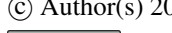



[Golovchanskaya and Maltsev, 2005]. But this condition can't be often satisfied. The
flapping wavelength can usually achieve 4 ~8 $R_E$, while the curvature radius of
magnetic field lines within the current sheet is only of 0.7-1.8 $R_E$. In addition, there
are often some transient events in the tail plasma sheet, such as bursty bulk flow
[Angelopoulos et al., 1992; Baumjohann et al., 1990; Cao et al., 2006,2013]. Some
current sheet flapping events were observed to be associated with these fast flows
[Sergeev et al, 2006; Duan et al., 2013]. Their flapping velocities are smaller (several
tens of kilometers per second) in the slow-flow regions.
Kink-like MHD waves have been considered as the generation mechanism of the
flapping motion [Daughton, 1999; Fruit et al., 2002, 2004]. Their phase speeds are
usually more than hundreds km/s and are not consistent with the flapping wave
propagating speeds of tens km/s [Sergeev et al., 2004]. Taking account of kinetic
effects, some models investigated drift K-H instabilities spreading with ion drift
speeds [Lapenta et al., 1997; Karimabadi et al., 2003]. Although these waves have
comparable propagation speeds, their propagation directions are unidirectional and
also don't match with the observations. Malova et al [2007] analyzed an asymmetric
current sheet model and suggested that hemispheric asymmetric plasma sources can
induce the flapping motions. Recently, Wei et al. [2015], using Cluster data,
demonstrated that current sheet flapping motions are closely related to periodic
hemispheric-asymmetric populations of bouncing particles. Their results imply that
the specific nonadiabatic ion behaviors inside the central plasma sheet may excite the
current sheet flapping motions.





In the sharp field reversal of the magnetotail current sheet, distinct classes of particle
orbits can be organized using the κ parameter introduced by Büchner and Zelenyi
[1986, 1989]. The κ parameter is defined as $\kappa = (R_c / \rho_i)^{1/2}$, where $R_c$ is the minimum
curvature radius of the magnetic lines and $\rho_i = (2kT_i/m_i)^{1/2}/\omega_i)$ is the maximum ion
gyroradius. At large (above ~3) κ values, the particles motion is adiabatic, and the
guiding center approximation is valid. For κ is between ~1 and ~3, particles behave in
a chaotic manner and their motion cannot be characterized by an invariant [Büchner
and Zelenyi, 1989]. For κ<1, those particles possibly meander inside the current sheet
[Speiser, 1965] and exhibit either transient, trapped, or quasi-trapped behaviors [Chen
and Palmadesso, 1986; Chen et al., 1990; Chen, 1992].
Furthermore, in the presence of a guiding field, the shear patterns of the magnetotail
current sheet in association with the non-adiabatic particle kinetic can be strongly
changed. Previous investigations on the non-adiabatic particles in the case of a
constant $B_y$ have been executed by both analytic methods and test particle simulations
[Zhu and Park, 1993; Delcourt et al., 1996; Malova et al., 2012; Grigorenko et al.,
2013]. These results revealed that asymmetrical particle scattering can take place in
the vicinity of the neutral line. Particularly, Delcourt and Martin [1994] suggested an
impulsive centrifugal force model to describe the nonadiabtic particle behaviors in the
current sheet. In the presences of a guiding field, the effect of a nonzero $B_y$ is
considered to impart a rotation of the impulse centrifugal force when particles cross
the neutral line [Delcourt et al., 1996]. It can be enhanced or attenuated when the
rotation occurs along (against) the particle gyromotion [Delcourt et al., 1996, 2000].




Hence, particles launching from the north part of the current sheet will behave
differently from that launching from the south part when they cross the central line.
Meanwhile, the studies of magnetic configurations with a non-constant $B_y$ and the
associated non-adiabatic particle behaviors to reveal the formation of two general
self-organized shear structures, symmetric and antisymmetric respectively [Malova et
al., 2015].
In addition, due to their distinct motion behaviors, different classes of particles within
the plasma sheet are supposed to have different capabilities of carrying cross-tail
current. Thus, the particle population variations of different classes can change the
current structure of the current sheet. For the adiabatic trapped ions, theoretically they
do not carry any net current because their orbits are closed. Owing to strongly curved
serpentine-like motions near the neutral plane, the quasi-adiabatic particles can
support local currents that are directed oppositely to the general cross-tail current
[Zelenyi et al., 2000, 2002a, b, 2003, 2011]. Thus, the existence of prominent
populations of quasi-adiabatic particles will result in a current sheet with a bifurcated
profile [Zelenyi et al., 2003]. The magnetic field of a bifurcated current sheet has a
plateau profile between the peaks of the enhanced magnetic field gradient [Sergeev et
al., 1993, 2003]. Bifurcated current sheets are often observed during substorms
[Sergeev et al., 1993; Hoshino et al., 1996; Asano, 2001; Asano et al, 2003; Runov et
al., 2003a]. In addition, bifurcated current sheets can exist around the X line, with a
flat current sheet in between [Runov et al., 2003b].





In this paper, we reveal the symmetric and antisymmetric shear structures associated
with the flapping current sheet. The alternating bending orientations of the guiding
field are exactly corresponding to alternating North-South asymmetries of the
bouncing ion population in the sheet center. In addition, we present the existence of a
substantial population with dawnward motion in the center of a bifurcated current
sheet, which is identified as the quasi-adiabatic particles.

**2. Observations**
2.1 flapping current sheet with antisymmetric shear structures: 03 Aug 2004 event and
03 Aug 2001 event
Cluster satellites once observed long-standing flapping motions on 3 August, 2004
when they traveled in the central current sheet. Those flapping motions were
measured without notable fast flows (for the details, see Wei's et al., (2015) (Fig.2 a,
d)). Figure 2 shows the measurements of the first five current sheet crossings during
the period of 06:30-07:05UT, within which each encounter of the neutral line is
marked by vertical dotted line. As revealed by pervious investigations [Runov et al.,
2005; Petrukovich et al., 2006, 2008], although the current sheet inclined significantly
in association with the flapping motions, the magnetic fields in the central current
sheet are nearly unchanged and mainly in the z-direction.
Fig.3 shows the energy and polar angle spectrograms of energetic ions, which were
measured by the CIS/HIA experiment [Rème et al., 2001]. The polar angle is the



angle between the particle movement direction and $Z_{GSM}$ axis, in which $\pm 90°$
represent the direction parallel and antiparallel to $Z_{GSM}$ axis, respectively. First, during
each crossing of the current sheet center, there are periodic variations of the ion flux
concentrated at $\pm 90°$ polar angle respectively. In the magnetotail configuration, this
observation means that the bouncing ion populations are North-South asymmetrical.
For example, at the first crossing of the current sheet center, the ion flux moving
upward is $\sim 8 \times 10^4$ count/spin, while the flux moving downward is $\sim 10^5$ count/spin.
Second, these North-South asymmetries of the ion fluxes are alternating between the
adjacent crossings of the sheet center. When the current sheet moved downward, the
ion fluxes moving northward are smaller than that moving southward. While when the
current sheet moved upward, the asymmetries are exactly in the opposite, that is, the
ion fluxes moving northward are larger than that moving southward. Third, as shown
in Fig.3c, there are polarization variations of $\gamma_{cy}$ (the y-component of the magnetic
field line curvature) measured at the current sheet center. The non-zero $\gamma_{cy}$ is a
manifestation of the bending of the magnetic field line to the y-direction developed in
the current sheet center in association with its flapping motion. The sign change of $\gamma_{cy}$
means an orientation change of the field line curvature y-component, where the field
line shape has an antisymmetric configuration (or in other words, an antisymmetric
shear structure, see the schematic Fig.1a or Figure 4b in Malova et al, 2015).
Although at the first look, all the polarization variations of $\gamma_{cy}$ seem to have similar
profiles, in fact, the polarization variations are alternating. For example at the first
crossing of the current sheet center, $\gamma_{cy}$ changes from negative to positive, where its



negative (positive) value corresponds to negative ( positive ) $B_x$ respectively.
However at the next crossing, although $\gamma_{cy}$ also changes from negative to positive, its
negative (positive) value corresponds to positive (negative) $B_x$ respectively. Thus, the
bending in the y-direction of the magnetic field lines measured at the first crossing is
actually opposite to the bending direction of the next crossing and so on for
subsequent crossings. Alternating orientations of the adjacent crossings are presented
as the solid line and the dashed line respectively in Fig. 1a. This picture of the
antisymmetric orientation changes of the magnetic field line curvature in the
y-direction can also be equivalently interpreted as an effective guiding field existing
near the current sheet center due to its alternating inclinations [Wei et al, 2015].
The particle motion in the magnetotail configuration is usually classified by its
adiabaticity parameter. In this event, the adiabaticity parameter $\kappa = (R_c / \rho_i)^{1/2} \sim 1.1$,
where $R_c \sim 2000$ km, is the curvature radius and $\rho_i \sim 1600$ km, is the ion gyroradius.
As mentioned above, the nonadiabtic particle behavior can be affected by the guiding
fields, which result in the asymmetrical particle scattering in the vicinity of the neutral
line. Especially, it is convenient to consider it as the existence of an impulse
centrifugal force due to the field line curvature y-component. Therefore, a series of
alternating hemispheric asymmetric ion populations can develop due to the
occurrence of the alternating orientations of the field line bending in the y-direction.
Theoretically, asymmetric plasma sources will violate the pressure balance condition
of the current sheet in the North-South direction and consequently drive the sheet to
reach a new equilibrium position [Malova et al., 2007]. Furthermore, this process is





self-consistent to preserve a complete flapping period. In the first half of a flapping
period, asymmetric plasma sources developed due to a specific orientation of the field
line bending and drove a vertical motion of the current sheet. Meanwhile, they reform
the shear configuration of the current sheet, i.e., the field line bending in the
y-direction, to the opposite orientation. In the second half of the flapping period,
opposite asymmetric plasma sources were caused by guiding fields with an opposite
bending orientation, and in their turn propel the current sheet to move vertically in the
opposite direction. Thus, the alternating asymmetric plasma sources maintain the
continuous oscillations of the current sheet. ( See the schematic Figure 1b in Wei et al,

228    2015).

A similar event of the current sheet flapping with antisymmetric shear structures is
shown in Figure 4. Two crossings of the neutral line during the period of
09:00-09:25UT on Aug 3 2001 were recorded. There are no high speed flows during
this current sheet flapping event (not showing in Fig.4). The ion flux moving upward
$(1.3*10^5 count/spin)$ is larger than that moving downward $(1.0*10^5 count/spin)$ at the
first travel through the neutral line, while the flux moving upward $(1.0*10^5 count/spin)$
are smaller than that moving downward $(1.4*10^5 count/spin)$ at the second travel
through. Corresponding to the alternating asymmetric ion fluxes, the alternating
orientations of the field line bending in the y-direction were measured as shown in Fig.
4c. In the first crossing of the current sheet center, the field line bending is to the
negative y-direction in the north-hemisphere and positive y-direction in the
south-hemisphere. While in the second crossing, the bending is to the positive
y-direction in the north-hemisphere and negative y-direction in the south-hemisphere.
In this event, the adiabaticity parameter $\kappa = (R_c / \rho_i)^{1/2} \sim 1.05$, where $R_c \sim 1000$km, and
$\rho_i \sim 900$km. As expected, $\kappa$ is within $\sim 1-3$ regime, so that the bouncing particles
behave in a chaotic manner in the vicinity of the neutral line and experience
asymmetric scattering. All the observational features can be interpreted as the same as
the above event.

2.2    Bifurcated flapping current sheet with symmetric shear structures: 26 Sept
2001 event
On 26 Sept 2001, during the period of 22:20-22:35 UT, Cluster observed a bifurcated
flapping current sheet event when they was located at [-18; 7; 0] $R_E$ (in GSM
coordinates), which had been previously reported by Sergeev et al [2003]. Figure 5
displays an overview of this event, in which no remarkable bulk burst flows were
encountered. As shown in Fig.5a, during each crossing of the sheet center, $B_x$ displays
a plateau profile, which is more easily identified during the last four crossings.
Correspondingly, the maximum cross-tail current $j_y$ deviates from the exact center of
the current sheet as shown in the third panel, that is, the total $j_y$ does indeed decrease
in the exact current sheet center and sometime displays a double-peak profile.
In association with the flapping motion of this bifurcated current sheet, alternating
hemispheric asymmetries of the bouncing ion population can as well be observed as
shown in Fig.6b. Corresponding to the hemispheric asymmetric of ion populations,
non-zero values of $\gamma_{cy}$ were observed during each crossing of the central current sheet.



Nevertheless, the measurements of the non-zero $\gamma_{cy}$ are different to that of the events
with antisymmetric shear structures. All these non-zero $\gamma_{cy}$ don't change their signs at
the crossing of the neutral plane, the non-zero $\gamma_{cy}$ without a sign change represents a
symmetric bending shape of the field lines in the y-direction (see the schematic Fig.
1b, or Figure 4a in Malova et al, 2015). Similarly, as shown as the solid and dashed
lines respectively in Fig.1b, the orientations of the bending shapes are alternating
between the adjacent crossings of the sheet center, that is, $\gamma_{cy}$ is positive when the
current sheet moved upward, while is negative when the current sheet moved
downward. Here, the adiabaticity parameter $\kappa = (R_c / \rho_i)^{1/2} \sim 0.9$, where $R_c \sim 500km$,
and $\rho_i \sim 540km$. Thus, asymmetric ion populations are supposed to develop due to the
alternating orientations of the bending shapes in the y-component.
Fig. 6c shows the azimuthal angle spectrograms of energetic ions.  The azimuthal
angle (-180° - + 180°) is the angle between the particle movement direction and X
axis, in which 0° represents the +x-direction, and ±90° represents the direction
parallel and antiparallel to y-axis respectively. During each crossing of the sheet
center except the first two, the angular distribution of a substantial population is
concentrate at -90°, which means that this population move to the -y-direction,
namely to the dawn direction. It is worth to note here that this particular observation
are rare and cannot be found in the above events (not shown in this paper). In this
event, the adiabaticity parameter $\kappa \sim 0.9 < 1$, thus it is reasonable to believe that a part
of the nonadiabatic ions belongs to the so called quasi-adiabatic particles. Those
particles meander in the current sheet center, and have a drift motion in the



-y-direction owing to their strongly curved serpentine-like motions near the neutral
plane. Thus, this population carries a net current opposite to the cross-tail current in
the sheet center. Consequently, the total cross-tail current carried by the whole
population will decrease in the sheet center and display a bifurcated profile, that is,
the current density maximum is not at the sheet center as the usual Harris-type sheet.
Here, the fact that there exists a substantial population with dawnward motion in a
bifurcated current sheet gives a firm observational evidence for its theoretical
generation mechanism.

2.3 Flapping current sheet with mixed shear structures: 26 Oct 2002 event
Bulk burst flows are common phenomena in the magnetotail. Figure 7 shows the
overview of a flapping current sheet with high speed flows observed by C3 on 26 Oct
2002. There are 8 crossings of the current sheet center during the period of
09:18-09:28UT. The periods of current sheet flapping are less than 2 minutes.
Apparent earthward fast flows were observed during this current sheet flapping event
as shown in Fig. 7d. The maximum high speed flow $V_x$ is larger than 400km/s, which
encountered at the seventh crossing. The burst feature of this flow can be found from
the observations that the flows encountered at the second and seventh crossings are
remarkable, while the ones encountered at the fourth and fifth crossings are ignorable.
In this flapping current sheet with high speed flows, the alternating hemispheric
asymmetries of the bouncing ion population can also be recognized as shown in
Fig.8b. However, the asymmetries recorded at the upward motions of the current sheet



seem to be more pronounced than the ones recorded at the downward motions, except
the last one. Meantime, the upward amplitudes of the current sheet flapping seem to
also be more pronounced than the downward amplitudes, except the last two flapping
motions. Two populations concentrated at 0° polar angle were encountered between
the second and third crossings and at the seventh crossing, which are the earthward
fast flows. Corresponding to the hemispheric asymmetric of ion populations, non-zero
values of $\gamma_{cy}$ were observed in the vicinity of the neutral line. Beside the non-zero $\gamma_{cy}$
with a sign change, the non-zero $\gamma_{cy}$ without a sign change was recorded at the second,
fourth and sixth crossings of the sheet center. Thus, the shear structures in this event
are mixed, i.e. both symmetric and antisymmetric bendings are existed. It is contrast
to the events presented above, where all the non-zero $\gamma_{cy}$ have similar profiles.    In
this event, the adiabaticity parameter $\kappa = (R_c / \rho_i)$ 1/2 ~1.8, where $R_c$ ~ 1000km, and
$\rho_i$ ~ 315km. Although the magnetic structures with fast flows are more complex than
that of the case without fast flows, the intrinsic excitation mechanism of flapping
motion induced by nonadiabatic ions are still prevailed.

**3. Discussion and conclusion**
In association with the current sheet flapping motion, there are two type bending
patterns of the field lines in the y-direction, symmetric and antisymmetric as revealed
by the observations. In 03 Aug 2004 and 03 Aug 2001 events, the unperturbed
magnetic fields in the current sheet center are mainly in the z-direction.
Correspondingly, their bending patterns are antisymmetric. This picture can be



equivalently interpreted as an effective guiding field due to the inclinations of the
current sheet itself (Wei et al, 2015). While in 26 Oct 2002 and 26 Sept 2001 events,
both the z-component and y-component of the unperturbed magnetic fields are rather
small. Correspondingly, their bending patterns are symmetric or complicated (two
patterns coexist). Thus, we infer that in the case of an unperturbed magnetic field with
a dominant z-component, it is favorable to develop an antisymmetric bending pattern
of the field lines in the y-direction during the flapping motion, while in the other case;
it is favorable to develop an antisymmetric bending pattern. It is consistent with the
conclusion drew by the previous investigations that a higher probability of formation
of symmetric shear configuration at lower values of the normal magnetic component
[Rong et al., 2011; Malova et al., 2015].
Although an understanding of the asymmetric particle scattering of the bouncing ion
populations by guiding fields has been pointed out in Wei et al (2015), the scenario
shown here relevant to the detailed shear structures of the current sheet is more
general since the two kinds of shear patterns are self-consistently formatted from
some initial magnetic perturbation [Malova et al., 2015]. Also, in the view of an
impulse centrifugal force model which is applicable to describe ion behaviors with
adiabaticity parameter $\kappa \sim$ 1-3 as in the case of flapping events, it is more convenient
to investigate directly the magnetic line curvature rather than an effective guiding
field.
In summary, observations of flapping current sheet in the magnetotail are presented to
reveal their intrinsic excitation mechanism induced by nonadiabatic ions. The current





sheet up-down motions are exactly corresponding to alternating hemispherical
asymmetries of the bouncing ion population. These asymmetric ion populations are
present in the magnetic field configuration with a local bending y-component and
interpreted as a result of the nonadiabatic particle scattering in the vicinity of the
neutral line. Hence, the alternating asymmetric ion populations can develop due to the
occurrence of the alternating orientations of the field line bending. Those alternating
asymmetric plasma sources consequently induce the current sheet flapping motion as
a driver. In addition, we present the observations that there exists a substantial
population with dawnward motion in the center of a bifurcated current sheet. This
population is identified as the quasi-adiabatic particles, which supports a net current
opposite to the cross-tail current. The present results suggest that nonadiabatic ions
play a substantial role to determine current sheet dynamics, both its bulk mechanical
instability and current profiles.











### Acknowledgements

This work is supported by the National Natural Science Foundation of China (NSFC) under Grant No. 41174144 and 40974098 and the Specialized Research Fund for State Key Laboratories. Cluster data used in this paper are available via the Cluster Science Archive (https://www.cosmos.esa.int/web/csa/).

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





respectively. The values of 0° azimuthal angle represents the +x-direction, and ±90°
represent the direction parallel and antiparallel to y-axis respectively. d) The magnetic
field curvature y-component. e) current components and f) magnetic components.
Fig7. Detections of the flapping motion with bulk flows on 26 October 2002. a) The
magnetic field in GSM coordinates. b) The density measurements. c) The current and
d) Plasma velocity.
Fig8. Ion observations of alternating hemispherical asymmetries of the bouncing ion
population on 26 October 2002. a and b) The omnidirectional energy and polar angle
spectrogram of hot ions, respectively. (c) The magnetic field curvature y-component.
d) current components and e) magnetic components.










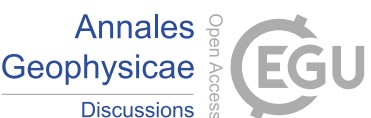

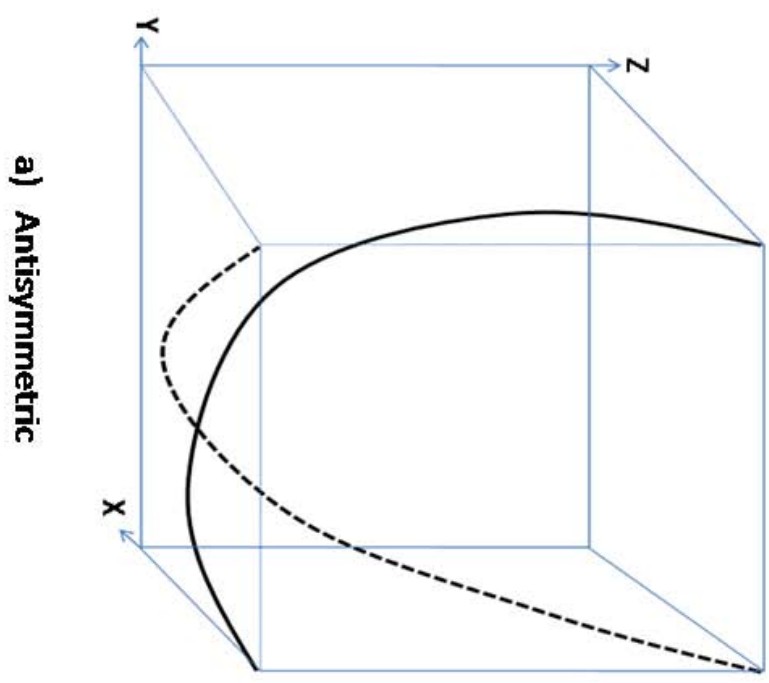

a) Antisymmetric

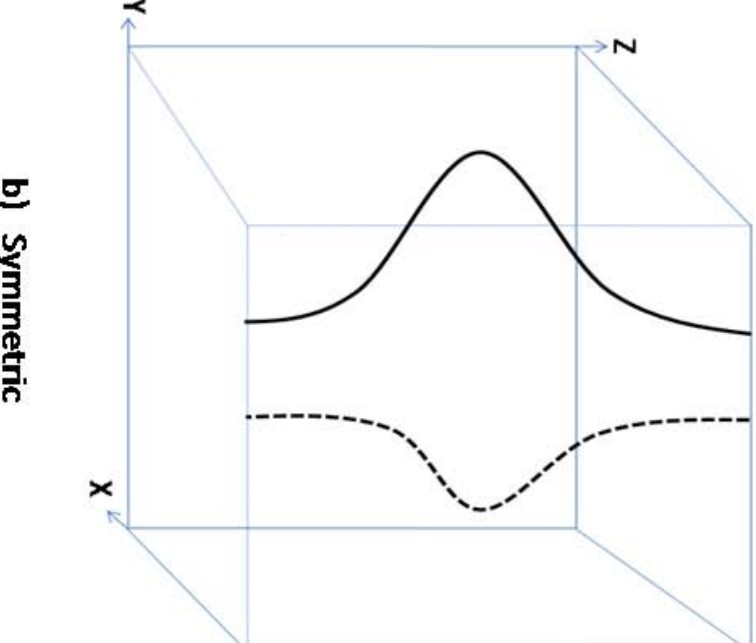

b) Symmetric











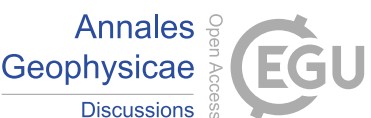















