# Peer review of "The current sheet flapping motions induced by non-adiabatic ions: case study"

_Annales Geophysicae, 2018_

## Referee Comment (RC1) · Anonymous Referee #1 · 28 Dec 2018

Review on "The current sheet flapping motion. . ."

First, I should note that this manuscript is the slightly revised version of draft that has been reviewed for possible publication in JGR (several months ago) and, as I understand, has been rejected. During that revision, there were set of important questions that Authors did not try to address. As the current version of the manuscript is almost identical to the previous (rejected from JGR) version, I assume Authors still need to consider following comments and questions:

(A) The theta distribution of ion fluxes (that is shown for quite broad energy range) can demonstrate a simple anisotropy induced by the current sheet motion (i.e., anisotropy related to change of the direction of plasma flow along the current sheet normal direction), and does not relate to a fine structure of a nonadiabatic ion velocity distribution.

[Figure]

Much more work is needed to prove that Authors indeed observe some nonadiabatic ions and to properly remove the effect of flapping motion from observed ion distributions. To make a final conclusion about presence of nonadiabatic ions, Authors have to show ion velocity distributions in the current sheet reference frame and these distributions should be similar to model predictions for nonadiabatic ions (see, e.g., ion distrobutions in Burkhart et al. 1992 doi:10.1029/92JA00495 or Sitnov et al. 2004 doi:10.1029/2003JA010123).

(B) Antisymmetric shear in the current sheet occurs due to field-aligned currents with a proper polarity and spatial distribution, i.e. Bx∼By relation does not guarantee any shear but can be due to flaring effect (a coordinate system rotation). To prove that there is an antisymmetric shear, Authors should reconstruct the local coordinate system (LMN) and plot Bm(Bl) hodograph.

(C) The symmetrical shear means that By maximizes at the Bx reversal. . . I do not see this effect in the shown three events.

(D) Note, there is no theory showing that the asymmetry of nonadiabatic ion sources can induce the flapping motion. Cited Malova et al. 2007 study describes the stationary asymmetric current sheet model and, to my best knowledge, there is no simulation showing that the flapping can appear in this model.

(E) The discussion about By effects on the noadiabatic ion motion is based on several publications by Delcourt et al., but all these papers (as well as many other studied related to this topic) deal with the constant By. . . whereas Authors show observations with By strong varying and reversing around equatorial plane.

(F) In the last event, the field Bz almost vanishes around the equatorial plane. . . how can one calculate kappa parameter for so small and fluctuating Bz? This is important to show that the curvature of magnetic field lines can be reliably estimates for such events.

---

## Referee Comment (RC2) · Anonymous Referee #2 · 1 Jan 2019

This is an observation-based study with claims that the authors analysed magnetic field line curvature and find two kinds of shear structures of the flapping current sheet (symmetric and anti-symmetric) in association with the so-called bouncing ion populations in the sheet center. Actually they show some data and provide some discussions of these plots, but no actual proofs of NS asymmetric plasma sources or related ion sub-populations were provided (see comm.1) and no deep analyses of current sheet structure have been done (see also comment 2). Also, there was no attempt to summarize quantitative theory predictions and compare them systematically and quantitatively with the observed parameters. I can not recommend this paper for the publication.

1. A trivial consequence of flapping plasma sheet is that , in the presence of up/dawn kink motions of the plasma sheet , an alternative up/down net flux of the plasma ions

exists, related to corresponding VZ component of the bulk flow. (Alternating convective VZ have been demonstrated many times since first Cluster studies of flapping phenomenon, e.g., Sergeev et el., 2003 etc). Neither in previous paper (Wei et al. GRL 2015) nor in the submitted paper, I was able to find any proofs that demonstrated ion flux asymmetry is related to smth else but VZ-related shifts of the ion distributions, or that it is related to some specific localized ion sub-population. No real analyses of distribution functions is provided, and the flux asymmetry effect is actually very weak (which is due to small VZ amplitude, I suppose). The authors have to demonstrate explicitly that a specific ion population exists on top of up/down convecting plasma sheet distributions, otherwise this is a mere speculation and discussion of non-existing things.

2. A success in the understanding often depends much on proper methods and proper choice of the reference system. This is often neglected throughout this study. The examples are:

(a) Considerable part of paper contains discussions of magnetic curvature variations. However both current and magnetic field (also curvature) are displayed in GSM coordinates, neglecting such things as the large tilts of current sheet normals (towards dusk or dawn,etc), tail flaring effects (with magnetic field planes diverging downtail) etc. LMN-type coordinate systems could be a better choice (but they may vary between subsequent neutral sheet crossings). GSM is not a proper reference system in such kind of analyses, the observed GSM variations are not easily interpreted, and they are hard to compare with any model predictions. (b). Previous studies of current carriers showed that protons are not typically the main current carriers, and that electric drifts are important players in this game (see e.g. Runov et al. AnnGeo 2005, Artemyev et al AnnGeo 2009, etc ). Therefore, it is important to analyze the ion distributions in the plasma frame. This was already discussed in the comment 1, and another example is your finding of ion population moving dawnward, which was exposed as specific new result of this study but really it can be related to the dawnward plasma convection in

this particular episode. Note also that theoretical models are always formulated in the plasma frame. —end review

---

## Referee Comment (RC3) · Anonymous Referee #3 · 9 Jan 2019

This article deals with important question of the sources of vertical flapping motions of relatively thin current sheet in the Earth's magnetotail. General interpretations of this phenomenon have been proposed by Sergeev et al. (2004) as kink-wave propagating from the magnetotail center toward its flanks. However, available interpretations can not explain all features of flapping waves. In this work authors tried to explain vertical current sheet oscillations by the asymmetric distribution of plasma population that can play the role of their driver. Thus the shape of magnetic shear component (symmetric and anti-symmetric cases are considered) is followed by the North-South asymmetry of ion distribution, that can be a source of flapping motion of current sheet. This result agrees with works by Malova et al., JGR, 2012, 2015 where it was shown that (1) magnetic shear component By, symmetric and antisymmetric relatively Z coordi-

nate in GSM system, can be self-organized in current sheet; (2) magnetic shear in current sheet can lead to the Northern-Southern asymmetry of plasma distributions in the vicinity of current sheet.

The general question of this investigation is the identification of quasi-adiabatic plasma population. Generally one can use two well known methods: 1) estimate of parameter of quasi-adiabaticity kappa (introduced by Buechner and Zelenyi, JGR, 1989) in current sheet neutral plane, that should be less than 1 for quasi-adiabatic particles; 2) investigation of {Vx, Vy} plasma velocity distribution, that should have characteristic "banana" shape in the neutral plane (such identification in the magnetotail was done, e.g., in (Artemyev et al., Ann. Geo., 27, 2009; JGR, 115, A12255, 2010) and theoretically predicted by Burkhart et al., JGR, 1992).

Since the kappa estimate is rather approximate (I leave aside the question raised by Referent#2 about the transition to the local coordinate system related with current sheet), one should supplement the article by figure with plasma velocity distribution {Vx,Vy}, which the authors presented in Fig.4 in their partial response to Referent#1 (and possibly other figures 1-3 with corresponding interpretations). This distribution has a pronounced "banana" shape, which indicates a quasi-adiabatic particle dynamics. Generally I think the general idea of this work is new and interesting. Authors presented quite convincing and reasonable results. I recommend this work for publication after inserting additional figures with appropriate explanations of the presence of quasi-adiabatic particles in current sheet.

Please also note the supplement to this comment:
https://www.ann-geophys-discuss.net/angeo-2018-124/angeo-2018-124-RC3-supplement.pdf

———————————

---

## Author Comment (AC1) · 9 Jan 2019

**Partial response to reviewer #1**

We thank the reviewer for his/her detailed review and comments. For the sake of an interactive discussion to clarify the N-S asymmetry problem, this partial response is dedicated to **Comment A**. A full response to all the comments and revision to the manuscript are preparing and will be presented later.

**Comment** (A) The theta distribution of ion fluxes (that is shown for quite broad energy range) can demonstrate a simple anisotropy induced by the current sheet motion (i.e., anisotropy related to change of the direction of plasma flow along the current sheet normal direction), and does not relate to a fine structure of a nonadiabatic ion velocity distribution. Much more work is needed to prove that Authors indeed observe some nonadiabatic ions and to properly remove the effect of flapping motion from observed ion distributions. To make a final conclusion about presence of nonadiabatic ions, Authors have to show ion velocity distributions in the current sheet reference frame and these distributions should be similar to model predictions for nonadiabatic ions (see, e.g., ion distrobutions in Burkhart et al. 1992 doi:10.1029/92JA00495 or Sitnov et al. 2004 doi:10.1029/2003JA010123).

**Response**: It seems that the explanation of the asymmetrical theta distribution by the reviewer can be shown as **Schematic 1A**, if we didn't misunderstand the reviewer's statement. In that case, original symmetrical distributions become asymmetrical between the left and right parts due to a reference frame shift (even small). The asymmetry will be more prominent in the smaller velocity and/or theta domain. However, the observational theta distribution is shown as **Schematic 1B** (see also **Fig.2** and **Fig.3** below). In this situation, symmetrical distributions in the larger velocity and theta domain (the yellow and blue parts) will maintain the original distributions, even though there is a rather small reference frame shift (<50km/s). The population in the smaller velocity and/or theta domain (gray part) is excluded automatically in the comparison of the symmetry between the theta distributions concentrated on $+90^0$ and $-90^0$. A slight distribution difference between the yellow and blue parts due to the frame shift is only in theory and is out of the instrument resolution.

**Fig.2** shows the ion distributions in the sheet bulk motion frame at the first sheet center crossing in the first event (03/08/2004). It can be seen that there is a distribution asymmetry between the top and bottom of the $V_z$-axis, which correspond to the blue and yellow part in **Schematic 1B** respectively. The asymmetric fluxes are $\sim 8\times 10^4$ vs. $\sim 10^5$ count/spin, as mentioned in the manuscript. As we clarify above and also the frame shift has already been taken into account, this asymmetry seems not to be

caused by the current sheet bulk motion.

To verify the asymmetry signature more clearly, **Fig.3** shows some theta-phi angular distributions in a single energy level (the x-axis is the phi angle from $-180^0$ to $+180^0$). The theta distribution asymmetries between $+90^0$ and $-90^0$ can be found in two higher energy levels, as shown in the two panels on the top. As a comparison, no asymmetry is displayed in lower energy levels, as shown in the two panels on the bottom. If it is interpreted by a frame shift according to the reviewer's explanation, the distributions in lower energy levels should also be asymmetrical. From the view of individual particles for these approximate monoenergy populations, the particle movement direction in the z-direction (~1500-2000km/s), both parallel and antiparallel, cannot be changed by a too small frame shift (<50km/s).

According to the reviewer's suggestion, we check some ion nonadiabatic signatures in the ion distribution in higher energy range (in the sheet bulk motion frame). An asymmetric profile of the $V_x$-$V_y$ distribution (18-32keV) is shown in **Fig.4**, which seems to be one of the ion nonadiabatic scattering features and similar to the fig.12a in Burkhart et al. 1992 (for larger adiabaticity parameter <1). Of course, it is a preliminary result on the ion non-adiabaticity. Still, the observational evidences reveal the existence of the N-S symmetrical populations, if the reviewer agrees with this point, which is a direct consequence of

nonadiabatic ions interacting with the current sheet. A complete identification of the ion nonadiabatic behaviors may be left to further investigations.

[Figure]

[Figure]

[Figure]

CIS-HIA

SAMBA (SC 3)

03/Aug/2004 06:38:35.339

[Figure]

Vy (km/s)

-2000

0

2000

-2000

0

2000

Vx (km/s)

Ions

Log c/s

3.7
3.3
2.9
2.5
2.1
1.8
1.4
1.0

---

## Author Comment (AC2) · 9 Jan 2019

The comment was uploaded in the form of a supplement:
https://www.ann-geophys-discuss.net/angeo-2018-124/angeo-2018-124-AC2-supplement.pdf

---

## Referee Comment (RC4) · Anonymous Referee #2 · 22 Jan 2019

This is a brief comment to your partial response. I don't think that by drawing simple two-color cartoons or showing some particular slices of distribution function you can prove that a specific ion population exists on top of up/down convecting plasma sheet distributions (VZ-shifted Maxwellian ot Kappa).

When analysing shifting Maxwellian for realistic temperature/density/flow one can understand that measurable angular asymmetry due to the flow appears in the high-energy part of spectrum, where the energy flux drop fast with the increasing energy. This was known for a long time, particularly Roelof et al JGR 1976 (https://doi.org/10.1029/JA081i013p02304) used the instrument, only capable to measure the ions above 50keV, to detect rather weak flows of 50km/s or comparable (this is due to rather soft proton spectra at those energies, so that a 20-50km/s shift of velocity

distribution causes a measurable angular anisotropy). No such effect would be seen at smaller energies near the peak of E-flux distribution near the thermal energy. Particularly, this explains your Phi/Theta plots. I believe this is a very probable explanation of your NS asymmetric particle flux plots.

Again, I reiterate that in order to demonstrate "a specific ion population existing on top of up/down convecting plasma sheet distributions" you need to work with the distribution functions. Ideally you have to show that a significant (asymmetric??) population remains after subtraction of the shifted Maxwellian/kappa (with realistic n,T,V) from the measured distribution. The error analyses should be important part of the story, you also may try E-field observations at Cluster to evaluate/confirm the true convective velocity. Without such analyses the paper is a discussion of non-existing phenomenon.
* * *

---

## Author Comment (AC3) · 22 Feb 2019

**Response to reviewer #3**

We thank the reviewer for the insightful comments and constructive suggestions. The responses and revisions to the manuscript are as following:

**Comment**: This article deals with important question of the sources of vertical flapping motions of relatively thin current sheet in the Earth's magnetotail. General interpretations of this phenomenon have been proposed by Sergeev et al. (2004) as kink-wave propagating from the magnetotail center toward its flanks. However, available interpretations cannot explain all features of flapping waves. In this work authors tried to explain vertical current sheet oscillations by the asymmetric distribution of plasma population that can play the role of their driver. Thus the shape of magnetic shear component (symmetric and anti-symmetric cases are considered) is followed by the North-South asymmetry of ion distribution, that can be a source of flapping motion of current sheet. This result agrees with works by Malova et al., JGR, 2012, 2015 where it was shown that

(1) magnetic shear component By, symmetric and antisymmetric relatively Z coordi- nate in GSM system, can be self-organized in current sheet; (2) magnetic shear in current sheet can lead to the Northern-Southern asymmetry of plasma distributions in the vicinity of

current sheet.

The general question of this investigation is the identification of quasi-adiabatic plasma population. Generally one can use two well known methods: 1) estimate of parameter of quasi-adiabaticity kappa (introduced by Buechner and Zelenyi, JGR, 1989) in current sheet neutral plane, that should be less than 1 for quasi-adiabatic particles; 2) investigation of {Vx, Vy} plasma velocity distribution, that should have characteristic "banana" shape in the neutral plane (such identification in the magnetotail was done, e.g., in (Artemyev et al., Ann. Geo., 27, 2009; JGR, 115, A12255, 2010) and theoretically predicted by Burkhart et al., JGR, 1992).

Since the kappa estimate is rather approximate (I leave aside the question raised by Referent#2 about the transition to the local coordinate system related with current sheet), one should supplement the article by figure with plasma velocity distribution {Vx,Vy}, which the authors presented in Fig.4 in their partial response to Referent#1 (and possibly other figures 1-3 with corresponding interpretations). This distribution has a pronounced "banana" shape, which indicates a quasi-adiabatic particle dynamics. Generally I think the general idea of this work is new and interesting. Authors presented quite convincing and reasonable results. I recommend this work for publication after inserting additional figures with appropriate explanations of the presence of quasi-adiabatic particles

in current sheet.

**Response**: As the reviewer pointed out, the kappa estimate is rather approximate and is more proper to be used as an auxiliary method to identify the nonadiabatic ion population. According to the reviewer's suggestion, we supplement a figure with plasma velocity distribution {Vx,Vy}, as shown in Fig.1

**Revision:** Fig.1 is inserted into the manuscript, and the corresponding interpretation is added to the text to describe the ion nonadiabatic signatures.

[Figure]

Vy (km/s)

Vx (km/s)

Ions

Log c/s

---

## Author Comment (AC4) · 22 Feb 2019

**Response to reviewer #2**

We thank the reviewer for his/her detailed review and comments. All the comments raised by the reviewer have been taken into account, and we hope that the reviewer will now find the paper acceptable for publication. The responses and revisions to the manuscript are as following:

**Comment 1** and **Response** see the **Partial response to reviewer #2**

**Comment 2**: A success in the understanding often depends much on proper methods and proper choice of the reference system. This is often neglected throughout this study. The examples are: (a) Considerable part of paper contains discussions of magnetic curvature variations. However both current and magnetic field (also curvature) are displayed in GSM coordinates, neglecting such things as the large tilts of current sheet normals (towards dusk or dawn,etc), tail flaring effects (with magnetic field planes diverging downtail) etc. LMN-type coordinate systems could be a better choice (but they may vary between subsequent neutral sheet crossings). GSM is not a proper reference system in such kind of analyses, the observed GSM variations are not easily interpreted, and they are hard to compare with any model predictions. (b). Previous studies of current carriers showed that protons are not typically the main current carriers, and that electric drifts are important players in this game (see e.g. Runov

et al. AnnGeo 2005, Artemyev et al AnnGeo 2009, etc ). Therefore, it is important to analyze the ion distributions in the plasma frame. This was already discussed in the comment 1, and another example is your finding of ion population moving dawnward, which was exposed as specific new result of this study but really it can be related to the dawnward plasma convection in this particular episode. Note also that theoretical models are always formulated in the plasma frame.

**Response**: The reason that field line curvature variations are displayed in GSM-system is to be consistent with the ion distributions since the guiding field effects on the asymmetric scattering of nonadiabatic ions can be more conveniently investigated. According to the reviewer's suggestion, we plot the guiding field hodograph in the LMN-system for the first event in the time interval 06:35-06:55UT. The change is especially noticeable in Bm, which reverses its sign between adjacent crossings.

The frame shift has already been taken into account to analyze the ion distributions (see also the **Partial response** to comment 1). As for the population moving dawnward, it is indeed exclusive to the bifurcated current sheet and exactly corresponds to the platform profiles in Bx in the sheet center. As one can see, comparable dusk-dawn plasma convections exist in all the presented events, while the population moving dawnward is unique to the bifurcated current sheet event. Thus, this population

seems not to be related to the dusk-dawn plasma convection.

**Revision**: The Bm-Bl hodograph Fig.1 is inserted as an additional figure to the manuscript.

**Comment 3**: This is a brief comment to your partial response. I don't think that by drawing simple two-color cartoons or showing some particular slices of distribution function you can prove that a specific ion population exists on top of up/down convecting plasma sheet distributions (VZ-shifted Maxwellian ot Kappa). When analysing shifting Maxwellian for realistic temperature/density/flow one can understand that measurable angular asymmetry due to the flow appears in the high-energy part of spectrum, where the energy flux drop fast with the increasing energy. This was known for a long time, particularly Roelof et al JGR 1976(https://doi.org/10.1029/JA081i013p02304) used the instrument, only capable to measure the ions above 50keV, to detect rather weak flows of 50km/s or comparable (this is due to rather soft proton spectra at those energies, so that a 20-50km/s shift of velocity distribution causes a measurable angular anisotropy). No such effect would be seen at smaller energies near the peak of E-flux distribution near the thermal energy. Particularly, this explains your Phi/Theta plots. I believe this is a very probable explanation of your NS asymmetric particle flux plots.

Again, I reiterate that in order to demonstrate "a specific ion population

existing on top of up/down convecting plasma sheet distributions" you need to work with the distribution functions. Ideally you have to show that a significant (asymmetric??) population remains after subtraction of the shifted Maxwellian/kappa (with realistic n,T,V) from the measured distribution. The error analyses should be important part of the story, you also may try E-field observations at Cluster to evaluate/confirm the true convective velocity. Without such analyses the paper is a discussion of non-existing phenomenon.

**Response**: In the first event, some plasma parameters are T~5600eV, $V_{zMax}$ ~ 40km. Thus, the anisotropy parameter $\alpha = V_{zMax} * V_{criteria} / V_{thermal}^{*2}$ ~ 0.04, where the $V_{criteria}$ ~ 1200km/s, is the criteria velocity of the population that displays asymmetric theta distribution, as can be seen in Fig.2 in the **Partial response**. The observational asymmetric profile $\ln(J(\theta=90^0)/J(J(\theta=0^0))$ ~$\ln(11000/7100)$ ~0.44. Thus, a theoretical anisotropy estimation using realistic plasma parameters is one order smaller than that of the observational profile.

---

## Author Comment (AC5)

**Response to reviewer #1**

We thank the reviewer for his/her detailed review and comments. All the comments raised by the reviewer have been taken into account, the responses and revisions to the manuscript are as following:

**Comment A** and **Response** see the **Partial response to reviewer #1**

**Revision:** The Fig.4 in the **Partial response** is inserted into the manuscript, and the corresponding interpretation is added to the text to describe the ion nonadiabatic signatures.

**Comment B**: Antisymmetric shear in the current sheet occurs due to field-aligned currents with a proper polarity and spatial distribution, i.e. Bx_By relation does not guarantee any shear but can be due to flaring effect (a coordinate system rotation). To prove that there is an antisymmetric shear, Authors should reconstruct the local coordinate system (LMN) and plot Bm(Bl) hodograph.

**Response**: According to the reviewer's suggestion, we plot the guiding field hodograph in the LMN-system for the first event in the time interval 06:35-06:55UT. The change is especially noticeable in Bm, which reverses its sign between adjacent crossings (Petrukovich et al. Ann. Geophys., 2006).

**Revision**: The Bm-Bl hodograph Fig.1 is inserted as an additional figure

to the manuscript.

**Comment C**: The symmetrical shear means that By maximizes at the Bx reversal: : : I do not see this effect in the shown three events.

**Response**: To the authors' understanding, it seems that in the symmetrical case By minimizes at the Bx reversal (see Fig.4a in Malova et al. JGR, 2015).

**Comment D**: Note, there is no theory showing that the asymmetry of nonadiabatic ion sources can induce the flapping motion. Cited Malova et al. 2007 study describes the stationary asymmetric current sheet model and, to my best knowledge, there is no simulation showing that the flapping can appear in this model.

**Response**: As the reviewer pointed out, in the Malova model the current sheet is stationary and generally cannot induce a flapping oscillation, in which the asymmetrical ion sources don't self-consistently interact with the current sheet. In the scenario described in this paper, the asymmetrical ion sources are locally generated and interact with the self-organized local shear structures (Malova et al. JGR, 2015). Thus, the asymmetries are alternating and can maintain a full flapping circle.

**Comment E**: The discussion about By effects on the noadiabatic ion

motion is based on several publications by Delcourt et al., but all these papers (as well as many other studied related to this topic) deal with the constant By: : : whereas Authors show observations with By strong varying and reversing around equatorial plane.

**Response**: As mentioned in the last response, nonadiabatic ions asymmetrically scattered by non-constant By and their self-consistent interaction with the current sheet was reported in Malova et al. 2015. To the authors' understanding, the non-constant By effects on the nonadiabatic ion motion is more general since the shear patterns are self-consistently formatted from some initial magnetic perturbation. Also, in the view of an impulse centrifugal force model which is applicable to describe ion behaviors with adiabaticity parameter $\kappa \sim$ 1-3 as in the case of flapping events, it is more convenient to investigate directly the magnetic line curvature rather than the guiding field itself.

**Comment F**: In the last event, the field Bz almost vanishes around the equatorial plane: : : how can one calculate kappa parameter for so small and fluctuating Bz? This is important to show that the curvature of magnetic field lines can be reliably estimates for such events.

**Response**: Theoretically, the estimation accuracy of the field line curvature depends on the characteristic scale of the Cluster spacecraft tetrahedron. When the curvature radii is larger than or comparable to the

tetrahedron characteristic scale, the estimation results can be regarded as reliable. It is satisfied in this event, where the curvature radii is ~2000-4000km and is comparable to the tetrahedron characteristic scale ~2000km. Although the kappa estimate is approximate, it is used as an auxiliary method to identify the nonadiabatic ion population since we also check the ion nonadiabatic signatures in the distribution functions (see the **Partial response**).